# Pilot-Scale Selective Electrodialysis for the Separation of Chloride and Sulphate from High-Salinity Wastewater

**DOI:** 10.3390/membranes12060610

**Published:** 2022-06-11

**Authors:** Fuqin Li, Yanfu Guo, Shaozhou Wang

**Affiliations:** School of Energy and Environmental Engineering, Hebei University of Engineering, Handan 056038, China; keyancaigou@163.com (Y.G.); wangshaozhou95@163.com (S.W.)

**Keywords:** high-salinity wastewater, monovalent selective electrodialysis, pilot-scale device, chloride and sulphate separation

## Abstract

The separation of chloride and sulphate is important for the treatment of high salt wastewater, and monovalent selective electrodialysis (MSED) has advantages in terms of energy consumption and pre-treatment costs compared to nanofiltration salt separation. Most of the research on monovalent anion-selective membranes (MASM) is still on a laboratory scale due to the preparation process, cost, and other reasons. In this study, a low-cost, easy-to-operate modification scheme was used to prepare MASM, which was applied to assemble a pilot-scale electrodialysis device to treat reverse osmosis concentrated water with a salt content of 4% to 5%. The results indicate that the optimum operating conditions for the device are: 250 L/h influent flow rate for the concentration and dilute compartments, 350 L/h influent flow rate for the electrode compartment and a constant voltage of 20 V. The separation effect of the pilot electrodialysis plant at optimal operating conditions was: the $Cl^{-}$ and $SO_{4}^{2-}$ transmission rates of 80% and 2.54% respectively, the separation efficiency (S) of 93.85% and the Energy consumption per unit of NaCl ($E_{NaCl}$) of 0.344 kWh/kg. The analysis of the variation of the three parameters of selective separation performance during electrodialysis indicates that the separation efficiency (S) is a suitable parameter for measuring the selective separation performance of the device compared to the monovalent selectivity coefficient ($P_{SO_{4}^{2-}}^{Cl^{-}}$).

## 1. Introduction

High-salinity wastewater is hard to treat and its direct discharge could cause serious harm to the natural environment; its main sources are: agricultural production (mainly irrigation drainage in areas with saline soils), industrial production (food processing, leather, petrochemicals, etc.), secondary sources (mainly from membrane and ion exchange technology) [1,2,3,4]. Sodium sulphate and sodium chloride are the main salts in high-salinity wastewater, and the removal of $Cl^{-}$ and $SO_{4}^{2-}$ is of a significant importance to the treatment of high-salinity wastewater. The conventional thermal crystallization process is energy intensive with insufficient product purity and difficult to recycle [5,6]. In response to the high energy consumption of conventional crystallization processes, a process that uses organic solvents to reduce the solubility of salts and precipitate the salts to achieve low energy consumption for salt removal has been applied on a large scale [7,8,9]. However, the products of the above treatment methods are often mixtures that are difficult to treat and their generation and storage can result in significant environmental risks [10]. Nanofiltration (NF) and monovalent selective electrodialysis (MSED) are widely used for the separation of mono-polyvalent ions from high- salinity wastewater to achieve product recovery.

NF is a widely used technique for salt separation. The mechanism of salt separation includes the Dornan effect, the sieving effect, the principle of dissolution diffusion and the conservation of charge [11,12,13], which has excellent effect on the separation of $Cl^{-}$ and $SO_{4}^{2-}$ in high-salinity wastewater. Yan et al. used nanofiltration (Desal-DL NF membrane) to treat a mixture of NaCl (23.4 g/L) and Na_2_SO_4_ (8.76 g/L), and obtained that the nanofiltration membrane had basically no effect on the retention of $Cl^{-}$, and the retention of $SO_{4}^{2-}$ was over 94% [14]. Pérez-González et al. used nanofiltration membranes (NF270, Dow Filmtec) to treat a mixture of $Cl^{-}$ (0.2–1.2 mol/L) and $SO_{4}^{2-}$ (0.1 mol/L), and obtained retention rates of 2–11% and 75–96% for $Cl^{-}$ and $SO_{4}^{2-}$, respectively [13]. However, the effectiveness of nanofiltration and membrane durability are significantly affected by membrane fouling, and the additional pretreatment process increases operating costs. Studies have pointed out that the chemicals used in the pre-treatment process can also be a new source of fouling and have a negative impact on the nanofiltration process [15,16,17,18]. According to relevant studies, selective electrodialysis has advantages in terms of salt separation, treatment cost and resistance to membrane fouling [19,20,21], and has broad application prospects.

The salt separation mechanism of selective electrodialysis mainly includes [22,23,24]: (1) pore-size sieving, as the hydration radius of $Cl^{-}$ (0.195 nm) is smaller than that of $SO_{4}^{2-}$ (0.300 nm) [25]. By enhancing the densities of the membranes or adding a dense modified layer to the membrane surface, the retention capacity of the membranes for multivalent ions can be improved and the mono-polyvalent selectivity can be improved. (2) Electrostatic repulsion, by creating electrically opposite charged layers on the surface of the ion exchange membrane, the higher valence ions are subjected to stronger electrostatic repulsion, increasing the retention capacity of the membrane for $SO_{4}^{2-}$. (3) Hydration energy difference, the hydration energy of $Cl^{-}$ and $SO_{4}^{2-}$ are -317 and -1000 kJ/mol respectively [26], the hydrophilicity of $SO_{4}^{2-}$ is more stronger than that of $Cl^{-}$, by reducing the hydrophilicity of the membrane, the transportation of $SO_{4}^{2-}$ can be hindered, which improves the monovalent selectivity [27].

Based on the above principles, Pan et al. prepared internally cross-linked monovalent selective ion exchange membranes using sulfadiazine as a cross-linking agent. The monovalent selectivity of the modified membranes rose due to the increased cross-linking and the introduction of the sulfonamide group. At optimal preparation conditions, the $P_{SO_{4}^{2-}}^{Cl^{-}}$ of the modified membrane for the $Cl^{-}$/$SO_{4}^{2-}$ system was 15.90, much stronger than the 1.28 for the non-sulfadiazine added membrane [28]. Liu et al. used layer assembly method to electrodeposit polymer on the surface of ion-exchange membrane and improved the stability of the modified layer by cross-linking agent. The introduction of the polymer and cross-linking agent enhanced the electrostatic repulsion and pore-size sieving, and the $P_{SO_{4}^{2-}}^{Cl^{-}}$ increased from 0.39 to 4.36 under the optimum modification conditions [29]. Liao et al. synthesized four alkyl spacers with different chain lengths and added them to the casting solution to prepare anion exchange membranes with different alkyl side chain lengths. The chain length of the alkyl spacers directly affected the hydrophilicity of the membranes and changed the monovalent selectivity. At the optimum preparation conditions, the monovalent selectivity coefficient $P_{SO_{4}^{2-}}^{Cl^{-}}$ was 7.10 [24]. 

However, research on monovalent anion-selective membranes (MASM) is still at the laboratory scale, and conventional MASM preparation schemes, such as the addition of crosslinkers [30,31], electrodeposition [32,33] and plasma [34,35], are complicated and costly to use for the preparation of large-scale MASM.

In this paper, a simple surface modification scheme was used to prepare a large scale MASM and the modified membrane was used to assemble a pilot-scale electrodialysis device with a commercial homogeneous cation exchange membrane. In the pilot-scale electrodialysis device, the area of the single ion exchange membrane is 800 cm^2^ (40 × 20 cm) and the total effective area of MASM is approximately 2 m^2^. The industrial high-salinity wastewater treated in the experiment came from the reverse osmosis concentrated water of a smelter. The main contaminants in the water were sodium sulphate and sodium chloride, with a salt content of 4% to 5%. The effects of influent flow, operation voltage and operation current on the separation effect of $Cl^{-}$ and $SO_{4}^{2-}$ were studied to determine the optimum operation conditions and calculate the relevant parameters.

## 2. Materials and Methods

### 2.1. Reverse Osmosis Concentrated Water

The main pollutants in the water are sulphate and sodium chloride, with a salt content of 4%–5% and a daily output of 80–90 m³. The main components of the reverse osmosis concentrated water are: about 22,000 mg/L of sulfate, about 15,000 mg/L of sodium ion, 20–70 mg/L of calcium ion, and about 7300 mg/L of chloride ion. The reverse osmosis concentrated water is softened by chemicals + ion exchange resin and then sent to the pilot-scale device for treatment.

### 2.2. Materials

Sodium poly(p-)styrene sulfonate (PSS, Mw = 70,000) was purchased from Shanghai McLean Biochemical Technology Co., Ltd., Shanghai, China.; 4,4′-Stilbene diazide-2,2′-disodium sulfonate tetrahydrate (DAS) was purchased from Aladdin Reagent (Shanghai) Co., Ltd., Shanghai, China; sodium chloride (NaCl), barium chloride (BaCl_2_), concentrated hydrochloric acid (HCl, 36%), silver nitrate (AgNO_3_), methyl red (C_15_H_15_N_3_O_2_), ethanolamine (HOCH_2_CH_2_NH_2_), potassium chromate (K_2_CrO_4_), anhydrous ethanol (CH_3_CH_2_OH, ≥99.7%), anhydrous sodium sulfate (Na_2_SO_4_), the above reagents are all analytically pure.

The anion exchange membrane (AEM) and cation exchange membrane (CEM) used in this experiment were purchased from Shandong Tian Wei Membrane Technology Co., Ltd., Shandong, China. The specific membrane performance parameters are shown in Table 1.

### 2.3. Modification Methods for AEM

The anion exchange membrane modification scheme was tested by static electrodialysis to determine the best modification scheme, and the effectiveness and durability of the modified membrane for the separation of $Cl^{-}$ and $SO_{4}^{2-}$ was determined by a small-scale electrodialysis experiment (total effective anion area of 1214.72 cm^2^).

The schematic diagram of the modification process of anion exchange membrane is shown in Figure 1. Cut the homogeneous anion exchange membrane into a 40 cm × 20 cm membrane to be modified, and rinse repeatedly to remove surface impurities; immerse the rinsed membrane to be modified into 1000 mL of a mixture of DAS (2.5 g/L) and PSS (1 g/L) for 6 h, and take it out after DAS, PSS and the surface layer material of the anion exchange membrane are fully combined; Immerse in 1000 mL deionized water, and immediately use UV light for 30 min (25 W UV lamps, 2 tubes were used in combination) to complete the cross-linking of the irradiated surface, record this surface as the preferential irradiated surface (side A), and then adjust the irradiated surface of the anion exchange membrane. The irradiated surface is exposed to ultraviolet light for 30 min, and the surface is recorded as the later irradiated surface (Side B); After the ultraviolet irradiation is completed, the anion exchange membrane is taken out, washed repeatedly to remove impurities that are not cross-linked in the membrane surface structure, and placed in a 2% Na_2_SO_4_ solution for later use. In use, the side A of the modified membrane faces the cathode of the device.

### 2.4. Pilot-Scale Electrodialysis Device

The pilot-scale electrodialysis device was manufactured by Hangzhou Lanran Technology Co., Ltd., Hangzhou, China., model SED2040-J001. The working cycle of the device is shown in Figure 2. The size of the membrane stack of the pilot-scale electrodialysis device is 40 × 20 cm, and the effective area of a single ion exchange membrane is 518.5 cm^2^ (30.5 × 17 cm). It consists of 40 anion exchange membranes and 41 cation exchange membranes. The total effective area of the AEM is 20,740 cm^2^. The volume of material in the circulation tanks of the Concentration, Dilute and Electrode liquid compartments is 10 L.

### 2.5. Determination of $Cl^{-}$ and $SO_{4}^{2-}$ Concentrations

The concentration of $Cl^{-}$ was determined by titration with silver nitrate and the concentration of $SO_{4}^{2-}$ was determined by weight method.

### 2.6. Measurement of Monovalent Anion Selectivity

#### 2.6.1. Monovalent Selectivity Coefficient $P_{SO_{4}^{2-}}^{Cl^{-}}$

The monovalent selectivity coefficient is calculated according to Equation (1)
(1)$$P_{SO_{4}^{2-}}^{Cl^{-}}=\frac{t_{Cl^{-}}/t_{SO_{4}^{2-}}}{c_{Cl^{-}}/c_{SO_{4}^{2-}}}=\frac{z_{Cl^{-}}\cdot J_{Cl^{-}}\cdot c_{SO_{4}^{2-}}}{z_{SO_{4}^{2-}}\cdot J_{SO_{4}^{2-}}\cdot c_{Cl^{-}}}$$
where $t_{i}$ is the transport number of the ions through the membrane; $z_{i}$ is the charge of the ions; $J_{i}$ is the flux of the ions (mol/m^2^·s) through the membrane; $c_{i}$ is the concentration of the ions (mol/L) in the diluate compartment.

The ions flux was obtained from the changes in concentration of the ions on the dilute according to Equation (2)
(2)$$J_{i}=\frac{V\cdot \frac{dc_{i}}{dt}}{A}$$
where *V* is the volume of the electrolyte solution in diluted compartment; *A* is the active area of the membranes.

#### 2.6.2. Separation Efficiency (S)

The separation efficiency is calculated according to Equation (3)
(3)$$S=\frac{\frac{c_{A}\left(t\right)}{c_{A}\left(0\right)}-\frac{c_{B}\left(t\right)}{c_{B}\left(0\right)}}{\left(1-\frac{c_{A}\left(t\right)}{c_{A}\left(0\right)}\right)+\left(1-\frac{c_{B}\left(t\right)}{c_{B}\left(0\right)}\right)}\times 100\%$$
where $c_{A}\left(t\right)$ and $c_{B}\left(t\right)$ are the concentrations of *A* ($SO_{4}^{2-}$) and *B* ($Cl^{-}$) in the diluate compartment; $c_{A}\left(0\right)$ and $c_{B}\left(0\right)$ are the initial concentrations.

#### 2.6.3. Ion Transport Number Ratio $n(Cl^{-}$/$SO_{4}^{2-})$

The ion transport number ratio is calculated according to Equation (4)
(4)$$n(Cl^{-}/SO_{4}^{2-})=\frac{c_{B}\left(0\right)-c_{B}\left(t\right)}{c_{A}\left(0\right)-c_{A}\left(t\right)}\times 100\%$$
where $c_{A}\left(t\right)$ and $c_{B}\left(t\right)$ are the concentrations of *A* ($SO_{4}^{2-}$) and *B* ($Cl^{-}$) in the diluate compartment, $c_{A}\left(0\right)$ and $c_{B}\left(0\right)$ are the initial concentrations.

### 2.7. Ion Transmission Rate η

The ion transmission rate is calculated according to Equation (5)
(5)$$\eta_{i}=\frac{c_{i0}-c_{it}}{c_{i0}}\times 100\%$$
where $c_{i0}$ is the initial concentration in the dilute compartment; $c_{it}$ is the concentration in the dilute compartment.

### 2.8. Energy Consumption per Unit of NaCl ($E_{NaCl}$)

The energy consumption for the output of 1 kg of NaCl in the concentration compartment is calculated according to Equation (6)
(6)$$E_{NaCl}=\int_{0}^{t}\frac{UIdt}{C_{t}\cdot V_{t}\cdot M}$$
where $E_{NaCl}$ is the energy consumption to produce 1 kg of NaCl in the concentration compartment, kW⋅h/kg; $C_{t}$ is the concentration of $Cl^{-}$ in the concentration compartment, mol/L; $V_{t}$ is the volume of the electrolyte solution in concentration compartment, L; *M* is the Molar mass of NaCl, g/mol.

## 3. Results and Discussion

### 3.1. Influence of Influent Flow on Electrodialysis Process

In this test, the influent flow of the concentration and dilution compartment were kept consistent, and the influent flow rates of the dilute and concentration compartment were adjusted to be 100 L/h, 150 L/h, 200 L/h, 250 L/h, and 300 L/h respectively, and the corresponding flow rate on the surface of the ion exchange membrane were 0.408 cm/s, 0.613 cm/s, 0.817 cm/s, 1.021 cm/s, and 1.225 cm/s, respectively. The pilot-scale electrodialysis device was controlled to operate under the condition of electrode compartment flow of 350 L/h and constant voltage of 20 V. When the device operated for 40 min, the transmittance of the device $Cl^{-}$/$SO_{4}^{2-}$ and the total anion transmittance rate with the flow are shown in Figure 3. The influent flow affects the unit energy consumption ($E_{NaCl}$) of the device and the effect of separation efficiency is shown in Figure 4.

It can be seen from Figure 3 that under the experimental conditions, the transmittance of $Cl^{-}$($\eta_{Cl^{-}}$) is significantly higher than the transmittance of $SO_{4}^{2-}$($\eta_{SO_{4}^{2-}}$), indicating that the device has excellent monovalent selectivity. This phenomenon is due to: (1) the ionic hydration radius of $SO_{4}^{2-}$ is larger than $Cl^{-}\;$ [25], which makes the resistance of $SO_{4}^{2-}$ passing through the ion exchange membrane larger than $Cl^{-}$; (2) the DAS-PSS modified layer contains a large number of sulfonic acid groups that can be dissociated at any pH, which makes the DAS-PSS modified layer have a strong negative charge and a stronger electrostatic repulsion to $SO_{4}^{2-}$ with a higher ionic valence, further increasing the resistance of $SO_{4}^{2-}$ to transit through the ion exchange membrane.

The effect of influent flow rate on the total anion transmission rate ($\eta_{ta}$) and the $\eta_{Cl^{-}}$ and $\eta_{SO_{4}^{2-}}$ were basically the same, which showed an overall trend of first increasing and then decreasing. When the influent flow rate was increased from 150 L/h to 200 L/h, the total anion transmission rate remained almost unchanged, while the $\eta_{SO_{4}^{2-}}$ increased from 5.00% to 5.99% and the $\eta_{Cl^{-}}$ decreased from 91.48% to 90.53%, indicating that the effect of increasing the influent flow rate on the $\eta_{SO_{4}^{2-}}$ was more significant in this flow rate range. The effect of increasing influent flow rate on the $\eta_{SO_{4}^{2-}}$ was more obvious in this flow range. The reason for the above phenomenon are: related studies [34,35,36] have shown that (1) $Cl^{-}$ has preferential transport in the electrodialysis module; (2) under the condition that the total anion concentration is the same, an increase in the percentage of $SO_{4}^{2-}$ will increase the resistance of the electrodialysis device, indicating that the electrical conductivity of $SO_{4}^{2-}$ is lower than $Cl^{-}$.The hydraulic residence time of the salt solution in the electrodialysis module gradually decreases as the influent flow rate increases. The charged ions are directed to transport within the electrodialysis device by the force of the electric field, with the same operating voltage, the number of ions that have transported gradually increases as the hydraulic residence time of the feed increases in the electrodialysis module, resulting in the electrical conductivity of the feed decreases, which increases the electrical resistance of the electrodialysis device. According to Equations (6) and (7), the output power of the power supply gradually decreases as the resistance of the feed liquid increases, and less energy is used for ion transportation under the same operation time conditions, which reduces the $\eta_{ta}$.
(7)$$P=UI=I^{2}R=\frac{U^{2}}{R}$$
(8)W=P⋅t

The total anion transmission rate curve in Figure 3 showed that at an influent flow rate of 100 L/h, the longer hydraulic residence time of the salt solution in the electrodialysis module resulted in a lower operating power of the device and a total anion transmission rate of 44.30%, which was less than the rest of the influent flow rate. In the range of 100~250 L/h, the total anion transmission rate of the device was relatively stable, with a maximum value of 47.50% at 250 L/h. At this time, the $\eta_{Cl^{-}}$ and $\eta_{SO_{4}^{2-}}$ were 91.71% and 5.44% respectively, and the device showed excellent monovalent separation. At an influent flow rate of 300 L/h, the $\eta_{ta}$, $\eta_{Cl^{-}}$ and $\eta_{SO_{4}^{2-}}$ of the device were significantly decreased, which was due to the fact that the hydraulic residence time of the salt solution in the electrodialysis module was too short and some ions flowed out of the module before transportation, resulting in a decrease in the number of ions completing transportation.

In Figure 4, the separation efficiency (S) of the device first decreases and then increases, which is related to the formula for the separation efficiency (Equation (3)), which characterizes the relative deviation of the ion transport rate [37], and an increase in $\eta_{SO_{4}^{2-}}$ reduces the value of the separation efficiency for a constant $\eta_{ta}$.

In conjunction with the data in Figure 3, the $\eta_{ta}$ of the device was 44.30% and 46.09% at the influent flow rate of 100 and 300 L/h respectively, with the $\eta_{ta}$ at a relatively low level, and the separation efficiency value was larger currently. The $\eta_{ta}$ of the device fluctuated within the range of 47.25% to 47.50% at the influent flow rate of 150–250 L/h, which made the separation efficiency value lower than that of the influent flow rate of 100 and 300 L/h. The $\eta_{Cl^{-}}$ and $\eta_{SO_{4}^{2-}}$ were 91.71% and 5.44% respectively at a influent flow rate of 250 L/h. The $\eta_{ta}$ reached a maximum of 47.50% and the separation efficiency was 88.79%. When the influent flow rate increased to 300 L/h, due to the short hydraulic residence time of the salt solution in the electrodialysis module, the $\eta_{Cl^{-}}$ and $\eta_{SO_{4}^{2-}}$ decreased compared with that at 250 L/h, which increased the separation efficiency value.

The reason for the above phenomenon is that due to the synergistic effect of the pore-size sieving effect and the electrostatic repulsion effect of the modified membrane, the transport resistance of $SO_{4}^{2-}$ is greater than $Cl^{-}$, resulting in the preferential migration of $Cl^{-}$ in the electrodialysis module. After the $Cl^{-}$ concentration of the salt solution in the electrodialysis module decreases to a certain level, $SO_{4}^{2-}$ is forced to transport to maintain the operation of the device, resulting in a decrease in the separation efficiency (S) of the device. Therefore, as the $\eta_{ta}$ of the device increases, the separation efficiency (S) gradually decreases. When the influent flow was 100 and 300 L/h, the $\eta_{ta}$ of the device was less than the rest of the influent flow, showing a higher separation efficiency. In the range of influent flow 150~250 L/h, the $\eta_{ta}$ of the device was relatively stable. When the influent flow of the device was 150 L/h and 250 L/h, the $\eta_{Cl^{-}}$ was over 91%, Energy consumption per unit of NaCl ($E_{NaCl}$) was at a low level. In practical applications, the treated water volume of 250 L/h is significantly higher than 150 L/h, and the $E_{NaCl}$ of the device at 250 L/h is at the lowest level. Therefore, the optimal influent flow rate of the pilot-scale electrodialysis device is 250 L/h.

### 3.2. Influence of Operation Voltage on the Electrodialysis Process

The influent flow of the concentration and dilution compartment of the pilot electrodialysis device was 250 L/h, and the flow rate of water into the electrode compartment was 350 L/h. The external voltage of the device was adjusted in order to make the device operate at a constant voltage of 10 V, 15 V, 20 V, 25 V and 30 V. Measured and calculated the operation time when the $Cl^{-}$ transmission rate was 80%, and the $SO_{4}^{2-}$ transmission rate, separation efficiency and Energy consumption per unit of NaCl ($E_{NaCl}$) under the operation time were analyzed. Figure 5 shows the effects of different voltages on the selected separation performance of the device, and Figure 6 shows the effects of different voltages on the operation time and $E_{\mathrm{NaCl}}$ of the device.

When the $\eta_{Cl^{-}}$ is the same, the increase of the $\eta_{SO_{4}^{2-}}$ will cause the decrease of the separation efficiency. The effect of voltage on $\eta_{SO_{4}^{2-}}$ was more obviously, in the voltage range of 10 V~25 V, the $\eta_{SO_{4}^{2-}}$ was negatively correlated with the increase of voltage, from 3.67% at 10 V to 2.47% at 25 V, after the voltage increased to 30 V, the $\eta_{SO_{4}^{2-}}$ increased slightly. The separation efficiency was influenced by $\eta_{Cl^{-}}$ and $\eta_{SO_{4}^{2-}}$. When the $Cl^{-}$ transmittance was a fixed value, the separation efficiency was determined by the $\eta_{SO_{4}^{2-}}$, and the trend was exactly opposite to the $\eta_{SO_{4}^{2-}}$. The separation efficiencies of the operating voltages of 20 V and 25 V are more satisfactory, 93.85% and 94.01% respectively.

The reason for the above phenomenon is that the voltage rises and the electric field force on the ions increases, which means that the driving force for the directional transport of the ions increases. When the voltage was not more than 15 V, the electric field force on $Cl^{-}$ and $SO_{4}^{2-}$ was relatively stable, and the $\eta_{SO_{4}^{2-}}$ decreased from 3.67% to 3.54% with the increase of voltage. In the range of voltage from 15 V to 25 V, the voltage increased, and the electric field force increased on $Cl^{-}$, and $Cl^{-}$ squeezed the channel of $SO_{4}^{2-}$ through the ion exchange membrane, which resulted in a significant decrease of $\eta_{SO_{4}^{2-}}$. When the voltage has exceeded 25 V, due to the higher negative charge of $SO_{4}^{2-}$, the influence of the enhanced electric field force was more obvious, and the electric field force of $SO_{4}^{2-}$ has overcome the resistance of the ion exchange membrane. Therefore, the $\eta_{SO_{4}^{2-}}$ increases, while the separation efficiency decreases.

The $E_{NaCl}$ with voltage reached 0.344 kWh/kg and 0.415 kWh/kg at operation voltages of 20 V and 25 V respectively. The operation time was shorter under the operation voltage of 25 V. The above phenomenon is caused by the redox reaction shown in Equations (8) and (9) in the electrode area of the electrodialysis module, where water molecules are electrolyzed into $O_{2}$ and $H_{2}$, this process consumes energy and the gas produced by the reaction escapes from the device, resulting in energy loss. Higher voltage increases the energy consumption of the device for electrolysis of water and the circuit, which raises $E_{NaCl}$. Therefore, the suitable voltage for the device is 20~25 V, which can be adjusted according to practical requirements. The optimum operation voltage for the unit under the test conditions in this paper is 20 V, which is the lower $E_{NaCl}$.
(9)$$2H_{2}O\to O_{2}↑+4H^{+}$$
(10)$$2H_{2}O\to H_{2}↑+2OH^{-}$$

### 3.3. Influence of Operation Current on the Electrodialysis Process

The influent flow of the concentration and dilution compartment of the pilot electrodialysis device was 250 L/h, and the flow rate of water into the electrode compartment was 350 L/h. The operation current of the device was adjusted to make the device operate at 3 A, 3.5 A, 4 A, 4.5 A and 5 A, corresponding to a current density of 5.786 mA/cm^2^, 6.750 mA/cm^2^, 7.715 mA/cm^2^, 8.679 mA/cm^2^ and 9.643 mA/cm^2^ respectively. The unit was operated at constant current, measured and calculated the operation time when the $Cl^{-}$ transmission rate is 80%, and the $SO_{4}^{2-}$ transmission rate, separation efficiency and Energy consumption per unit of NaCl ($E_{NaCl}$) under the operation time were analyzed. Figure 7 shows the effect of current on the separation performance of the device selection, Figure 8 shows the effect of current on the operation time and $E_{NaCl}$ of the device.

It can be seen from Figure 7 that the $\eta_{SO_{4}^{2-}}$ gradually decreases in the range of operating current 3.0–4.0 A, with a minimum value of 2.85% achieved at operation current 4 A, at which time the separation efficiency is 93.13%. The separation efficiency is influenced by $\eta_{Cl^{-}}$ and $\eta_{SO_{4}^{2-}}$. When the $\eta_{Cl^{-}}$ is a constant value, the separation efficiency is determined by the $\eta_{SO_{4}^{2-}}$, and the change trend is completely opposite to the $\eta_{SO_{4}^{2-}}$. When the operation current is lower than 4 A, increased current can significantly reduce the $\eta_{SO_{4}^{2-}}$, the $\eta_{SO_{4}^{2-}}$ gradually decreased from 5.24% at 3 A to 2.85% at 4 A, and the separation efficiency increased from 87.70% to 93.13%; after the operation current is higher than 4 A, the $\eta_{SO_{4}^{2-}}$ gradually increased to 3.56% at operation current of 5 A, and the separation efficiency decreased to 91.48%, slightly higher than 91.38% at 3.5 A. 

The reason for the above phenomenon is that the electric field force on the ion is enhanced by the higher current, which means that the driving force for the directional transport of the ions is enhanced. When the current is not greater than 4 A, the effect of the enhanced electric field force on $Cl^{-}$ is more obviously, $Cl^{-}$ squeezes the transmission channel of $SO_{4}^{2-}$, which makes the $\eta_{SO_{4}^{2-}}$ decrease significantly. With operation current above 4 A, the effect of enhanced electric field force is more obvious by the higher negative charge of $SO_{4}^{2-}$, which leads to an increase in the $\eta_{SO_{4}^{2-}}$.

The trend of the $E_{NaCl}$ and operation time were essentially the same as that of constant voltage operation. Because of the increased current, the energy consumption of the circuit and electrodes, two components that are not used for ion transport, which increased the value of $E_{NaCl}$. Combined with the data in Figure 7, the separation efficiency of the pilot-scale electrodialysis device is highest when operating at a constant current of 4 A and the $E_{NaCl}$ is at a low level. Therefore, the optimum operation current for the device is 4 A. Under these operation conditions, the separation efficiency was 93.13%, the $E_{NaCl}$ was 0.321 kWh/kg and the operation time was 28.70 min.

### 3.4. Comparison of Operation Conditions for Electrodialysis Device

Based on the experimental results in Section 3.2 and Section 3.3 of this paper, the $Cl^{-}$ and $SO_{4}^{2-}$ transmission rates, separation efficiency (S), and energy consumption per unit of NaCl ($E_{NaCl}$) of the device are relatively similar when the device is operated at a constant voltage of 20 V and a constant current of 4 A. To determine the best operation conditions of the device, the power and energy consumption of the above-mentioned two operating conditions are compared. Figure 9 shows the variation in power during the operation of the electrodialysis device, and Figure 10 shows a comparison of the energy consumptions of the electrodialysis device under different operation conditions.

From Figure 9, it can be seen that the power variations of the device at a constant voltage of 20 V and a constant current of 4 A are exactly opposite to each other. This phenomenon was caused by the change in the resistance of the electrodialysis module during operation. The initial conditions for the electrodialysis unit were: high-salinity wastewater for the dilute compartment feed and deionized water for the concentration compartment feed. There were an insufficient number of ions in the concentration compartment and inadequate electrical conductivity when the electrodialysis unit initially started operating, which left the overall resistance of the module at a high level. As the electrodialysis process was performed, the concentration of ions in the concentration compartment increased, causing the overall resistance of the module to decrease. Because of the directional transport of the ions, the ion concentration in the dilute compartment is always reduced during the electrodialysis process, which causes a reduction in the conductivity of the dilute compartment, and when the dilute compartment concentration dropped below a certain level, the resistance of the dilute compartment became the control condition of the electrodialysis module. Therefore, the resistance of the device decreased and then increased during the electrodialysis process. According to Equation (6), the trend of the change in the power curve of the device under constant voltage and constant current operation conditions is consistent with the change in the resistance of the device. In the electrodialysis process, the power range of the device at a constant voltage and constant current is 38 to 94 W and 60.4 to 134 W, respectively. Compared to constant voltage operation, constant current operation at the initial and final stages of the electrodialysis process could cause the device to operate at high power conditions due to the higher resistance, which may cause negative effects on the device.

From Figure 10, the energy consumption for constant current operation is above that for constant voltage operation during 0 to 15 min and 30.15 to 40 min, and the reverse is true for 15 to 30.15 min. According to Equation (7), combined with the data in Figure 9, the trend of change in the energy consumption of the device corresponds to the change in the resistance of the device. According to 3.2 and 3.3 of this paper, the $\eta_{SO_{4}^{2-}}$, separation efficiency (S) and $E_{NaCl}$ at constant voltage of 20 V and constant current of 4 A were 2.54%, 2.85%; 93.85%, 93.13%; 0.344 kWh/kg, 0.321 kWh/kg respectively for $\eta_{Cl^{-}}$ of 80%. The energy consumption per unit for constant current operation is better than constant voltage, but the $\eta_{SO_{4}^{2-}}$ is higher than for constant voltage 20 V operation, which results in lower separation efficiency. Combined with the data in Figure 9 and Figure 10, the power of constant current operation is significantly more than constant voltage operation after significant ion transport, which is not conducive to controlling the extent of the reaction and could have a detrimental effect on the device. Therefore, in the test conditions of this paper, the optimum operating conditions for the device are 250 L/h influent flow rate for the concentration and dilution compartment, 350 L/h influent flow rate for the electrode compartment, the constant voltage is 20 V.

### 3.5. Selective Separation Performance Changes during Electrodialysis 

As a result of pore-size sieving and electrostatic repulsion, the modified membrane has a higher resistance to $SO_{4}^{2-}$, which makes the $Cl^{-}$ preferentially transported. The decrease in the concentration of $Cl^{-}$ in the salt solution causes more $SO_{4}^{2-}$ to transport, changing the relevant parameter that measures the monovalent selectivity of the device. To determine the applicability and accuracy of each parameter, the monovalent selectivity coefficient ($P_{SO_{4}^{2-}}^{Cl^{-}}$), separation efficiency (S), and ion transport ratio $n(Cl^{-}$/$SO_{4}^{2-})$ were compared for different $\eta_{Cl^{-}}$ values. A pilot electrodialysis test was carried out to analyze the process under the operating conditions where the $\eta_{Cl^{-}}$ exceeded 95%. The test conditions are as follows: the influent flow of the concentration and dilution compartments of the pilot electrodialysis device was 250 L/h, and the influent flow of the electrode compartment was 350 L/h at a constant current of 4 A. Figure 11 shows the variation of the monovalent selectivity coefficient ($P_{SO_{4}^{2-}}^{Cl^{-}}$), separation efficiency (S), and ion transport ratio $n(Cl^{-}$/$SO_{4}^{2-})$ during the electrodialysis process.

The ion transport ratio $n(Cl^{-}$/$SO_{4}^{2-})$ is the most intuitive reflection of the selective separation performance of the device, while the higher values of $n(Cl^{-}$/$SO_{4}^{2-})$ indicate the stronger monovalent selectivity of the device. However, $n(Cl^{-}$/$SO_{4}^{2-})$ is difficult to measure under different operation conditions; therefore, the monovalent selectivity coefficient and separation efficiency are often used to analyze the monovalent selectivity of the device. It can be seen from Figure 11 that $n(Cl^{-}$/$SO_{4}^{2-})$ tended to decrease—it did so from 80.07 at 50% of $\eta_{Cl^{-}}$ to 10.74 at 95% of $\eta_{Cl^{-}}$. This was especially true when the $\eta_{Cl^{-}}$ increased from 60% to 65%, at which point $n(Cl^{-}$/$SO_{4}^{2-})$ showed a dramatic decrease from 65.38 to 45.24; the change in value was significant. The above phenomenon is due to the insufficient amount of $Cl^{-}$ in the desalination chamber liquid after a large amount of $Cl^{-}$ is transported, forcing more $SO_{4}^{2-}$ to complete the transport to maintain the operation of the device. Therefore, as the amount of $Cl^{-}$ completing transport increases, more $SO_{4}^{2-}$ is transported, which results in a decreasing trend of $n(Cl^{-}$/$SO_{4}^{2-})$.

From Figure 11, the $P_{SO_{4}^{2-}}^{Cl^{-}}$ changed basically the same trend as $n(Cl^{-}$/$SO_{4}^{2-})$ when the $\eta_{Cl^{-}}$ was not greater than 70%, especially when the $\eta_{Cl^{-}}$ increased from 60% to 65%, the $P_{SO_{4}^{2-}}^{Cl^{-}}$ decreased rapidly from 85.23 to 67.05, accurately reflected the change of device selectivity. When the $\eta_{Cl^{-}}$ was more than 70%, the $P_{SO_{4}^{2-}}^{Cl^{-}}$ began to show an increasing trend and increased rapidly with the improvement of the $\eta_{Cl^{-}}$. When the $\eta_{Cl^{-}}$ was 95%, the $P_{SO_{4}^{2-}}^{Cl^{-}}$ was 103.52, which phenomenon clearly does not correspond to the variation of the separation performance of the device. The reason for the above phenomenon is that from the Equation of the monovalent selectivity coefficient $P_{SO_{4}^{2-}}^{Cl^{-}}$ (Equation (1)), the concentration of $Cl^{-}$ in real time in the dilute compartment salt solution participates in the calculation as part of the denominator in the formula, and after the $\eta_{Cl^{-}}$ has exceeded 70%, the real-time concentration of $Cl^{-}$ in the light chamber salt solution is lower, which amplifies the $P_{SO_{4}^{2-}}^{Cl^{-}}$ after the formula.

The separation efficiency is not as accurate as the monovalent selectivity factor when the $\eta_{Cl^{-}}$ is not greater than 70%, and the change trend when the $\eta_{Cl^{-}}$ increases from 60% to 65% is not reflected in its curve, but the change trend of the separation efficiency is basically the same as $n(Cl^{-}$/$SO_{4}^{2-})$, which can reflect the actual situation of the separation performance of the device during the whole electrodialysis process more accurately. Therefore, the separation efficiency is more suitable as a parameter to measure the monovalent separation performance of the device.

## 4. Conclusions

(1) The optimum operation conditions for the pilot-scale electrodialysis device were: 250 L/h for the concentrated chamber and the light chamber, 350 L/h for the polar chamber, and a constant voltage of 20 V. The device under this condition achieved $\eta_{Cl^{-}}$ of 80%, the $\eta_{SO_{4}^{2-}}$ was 2.54%, the separation efficiency was 93.85%, the $E_{NaCl}$ was 0.344 kWh/kg and the operation time was 30.68 min.

(2) In the electrodialysis process, the monovalent selectivity coefficient ($P_{SO_{4}^{2-}}^{Cl^{-}}$) was more accurate than the separation efficiency when the $\eta_{Cl^{-}}$ was less than 70%, and after the $\eta_{Cl^{-}}$ exceeded 70%, $P_{SO_{4}^{2-}}^{Cl^{-}}$ would not be able to reflect the actual separation performance of the device. Therefore, the separation efficiency (S) is more suitable as a parameter to measure the separation performance of the device selection.

## Figures and Tables

**Figure 1 membranes-12-00610-f001:**
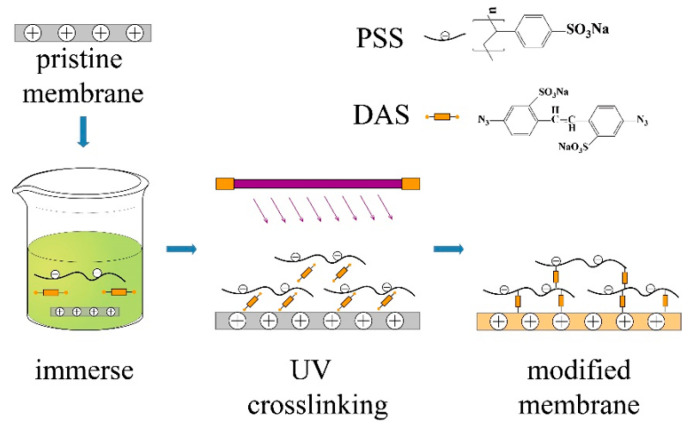
Diagram of the membrane modification process.

**Figure 2 membranes-12-00610-f002:**
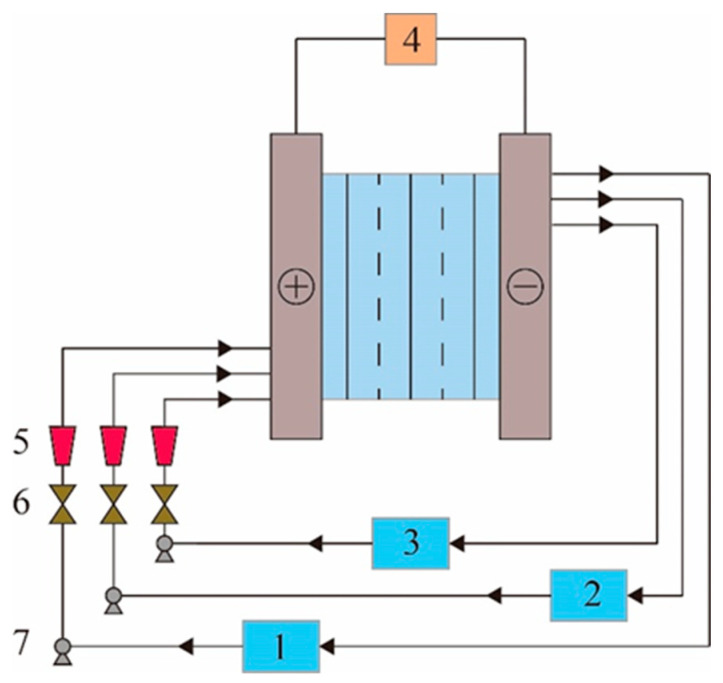
Schematic diagram of Pilot-Scale electrodialysis device. 1—Concentration compartment circulation tank; 2—Dilute compartment circulating tank; 3—Electrode liquid circulation tank; 4—DC stabilized power supply; 5—Flowmeter; 6—valve; 7—circulating pump.

**Figure 3 membranes-12-00610-f003:**
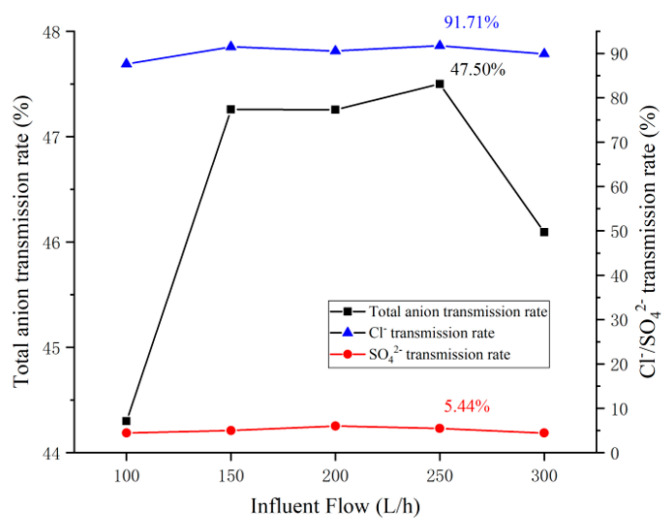
Influence of influent flow rate on the anion transmission rate of the device.

**Figure 4 membranes-12-00610-f004:**
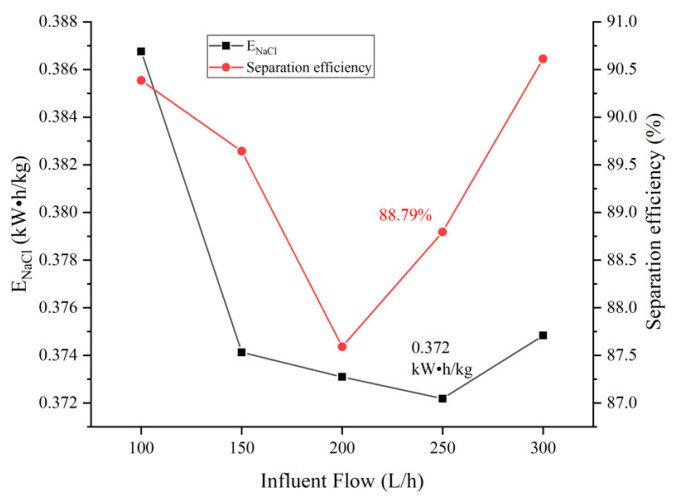
Influence of influent flow on the operating status of the device.

**Figure 5 membranes-12-00610-f005:**
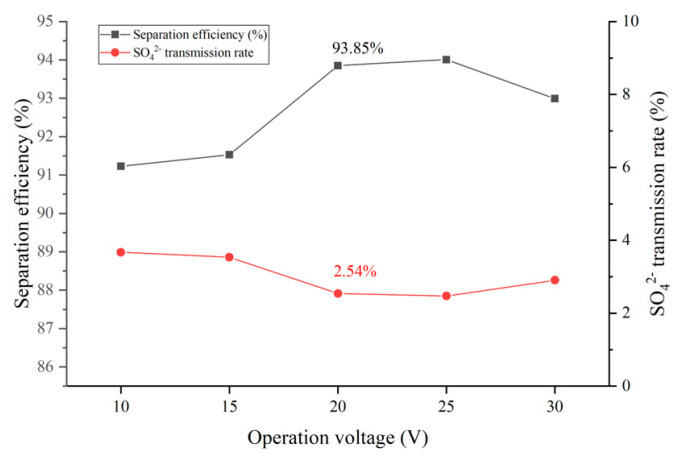
Influence of voltage on selection separation performance of the device.

**Figure 6 membranes-12-00610-f006:**
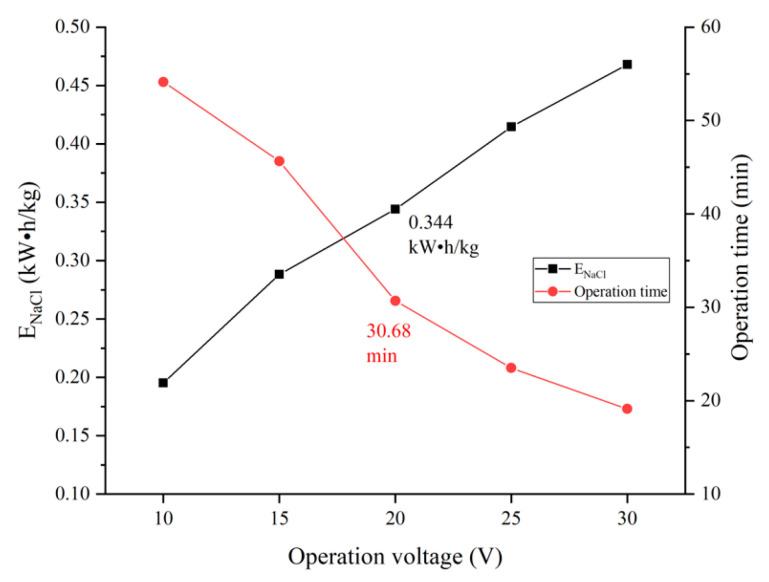
Influence of voltage on selection separation performance of the device.

**Figure 7 membranes-12-00610-f007:**
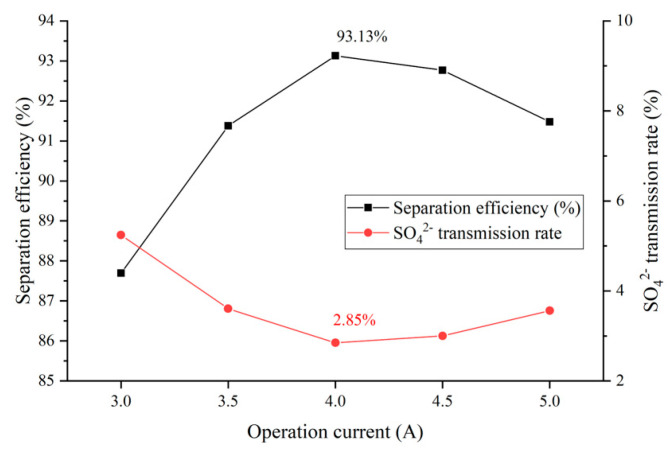
Influence of current on selection separation performance of the device.

**Figure 8 membranes-12-00610-f008:**
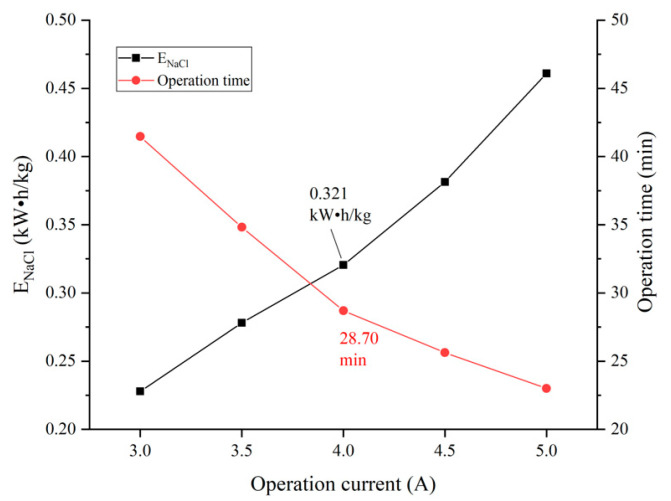
Influence of current on operating state of the device.

**Figure 9 membranes-12-00610-f009:**
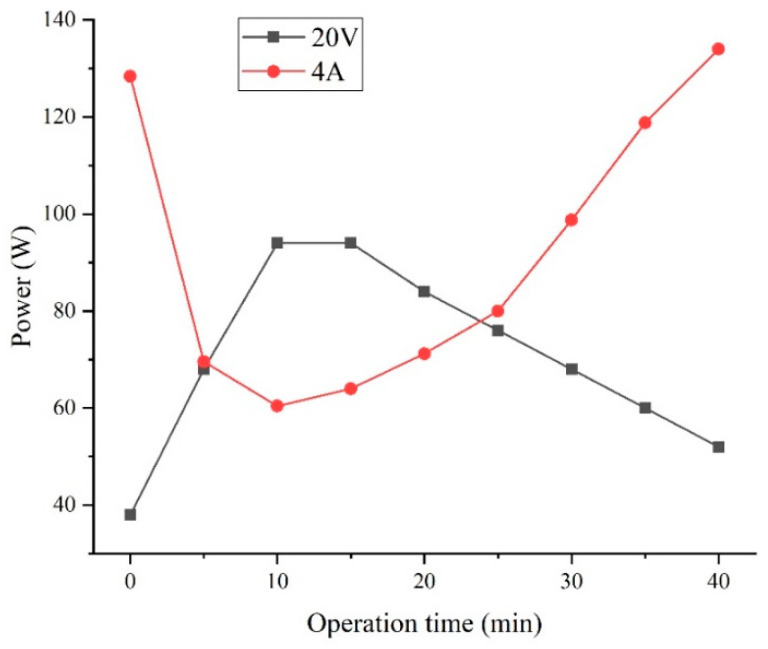
Power of the device under different operation conditions.

**Figure 10 membranes-12-00610-f010:**
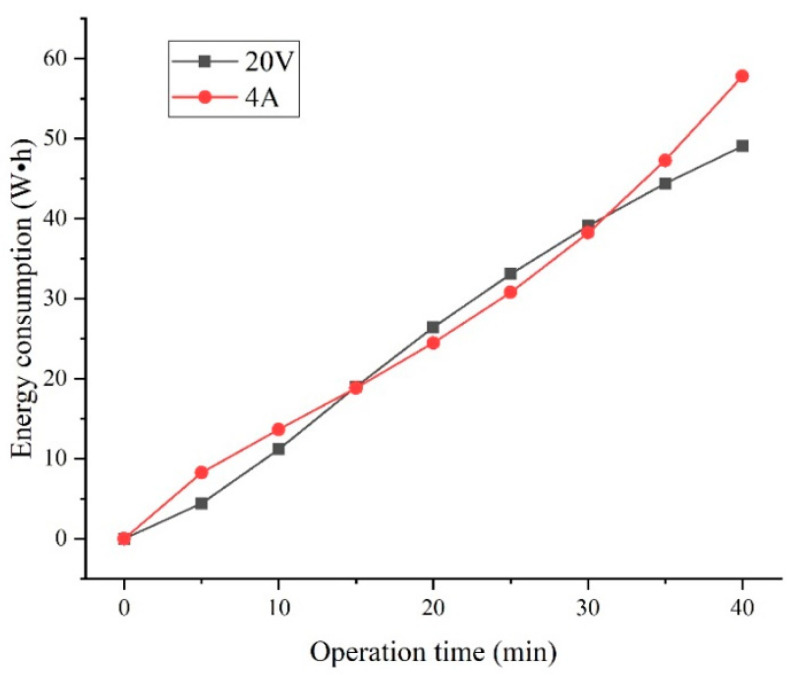
Energy consumption of the device under different operation conditions.

**Figure 11 membranes-12-00610-f011:**
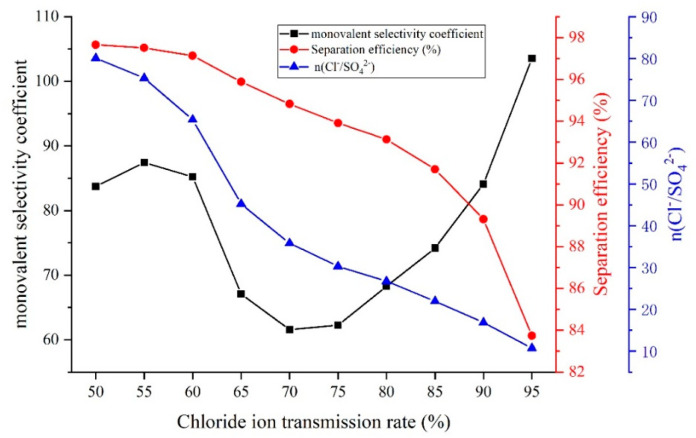
Trends of each index in electrodialysis process.

**Table 1 membranes-12-00610-t001:** Main performance parameters of ion exchange membranes.

Membrane Type	Model	Thickness in Wet Condition (mm)	Water Uptake (%)	$\mathrm{Electrical}\;\mathrm{Area}\;\mathrm{Resistance}\;(\Omega \cdot cm^{2},\;0.5\;M\;\mathrm{NaCl})$	Ion Mobility Number (0.5 M/0.1 M NaCl, 25 °C)
AEM	EDAIS-70nw	0.13–0.16	30–40	≤4	≥0.98
CEM	EDCIS-70nw	0.10–0.13	20–30	≤4	≥0.97

## Data Availability

Not applicable.

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
