# Peer review of "Pilot-Scale Selective Electrodialysis for the Separation of Chloride and Sulphate from High-Salinity Wastewater"

_membranes, 2022, doi:10.3390/membranes12060610_

Round 1

Reviewer 1 Report

The presented work provides information which is important for promoting water desalination processes based on membranes and also ED method, thus I recommend a minor revision prior to a possible publication in Membranes 

I have a few corrections and one suggestion below. 

Page 3, please correct the nomenclature in subparagraph Materials

please correct the description of units, for example as on page 3: 25w on 25 W in the subparagraph 2.3. Modification methods for AEM and correction 

please explain in this manuscript, how was calculated/measured/determined the degree of membrane surface membrane modification and the repeatability of the described method, otherwise, the reproducibility of the obtained results is uncertain.

Author Response

Point 1: Page 3, please correct the nomenclature in subparagraph Materials

please correct the description of units, for example as on page 3: 25w on 25 W in the subparagraph 2.3. Modification methods for AEM and correction

Response 1: Please provide your response for Point 1. (in red)

The relevant units, proper names have been amended. For example

"4,4-Stilbene diazide-2,2-disodium sulfonate tetrahydrate" to "4,4,-Stilbene diazide-2,2,-disodium sulfonate tetrahydrate".

"25w" to "25W"

Point 2: please explain in this manuscript, how was calculated/measured/determined the degree of membrane surface membrane modification and the repeatability of the described method, otherwise, the reproducibility of the obtained results is uncertain.

Response 2: Please provide your response for Point 2. (in red)

  1. Add the following description to the text:

The anion exchange membrane modification scheme was tested by static electrodialysis to determine the best modification scheme, and the effectiveness and durability of the modified membrane for the separation of  and  was determined by a small-scale electrodialysis experiment (total effective anion area of 1214.72cm2).

  1. A static electrodialysis unit as shown in Figures 1 and 2 was used to determine the optimum modification scheme using the univalent selectivity factor as an evaluation indicator.
  2. A small-scale electrodialysis experiment was conducted using a dynamic electrodialysis unit as shown in Figure 3 to determine the performance and durability of the modified membrane separation and to provide a reference for the operating parameters of the pilot-scale electrodialysis device.
  3. Figure 4(a) shows the pristine membrane and Figure 4(b) shows the modified membrane. The pristine membrane changed to brownish red after modification and the orange part of the membrane edge is caused by the insufficient reaction of the modifying agent, which is due to the reaction of the agent with the membrane surface being hindered by the fixation during the modification process. Therefore, the effect of the modification can be determined by observing the change in colour of the membrane surface.

Reviewer 2 Report

The scopes & findings of the study are worth to be published. However,  minor modifications are required as follows:

1. All result diagrams have to be revised particularly to highlight the optimum setting. Please refer the comments in the article. 

2. If possible, please discuss the findings in compared to the optimal settings from other pilot studies. 

3. At the pilot scale operation, economic analysis is a critical factor that has to be reflected upon. Perhaps, conduct brief OPEX analysis on the system, should it be running based on a continuous mode!  

Author Response

Responses to questions need to include relevant data graphs, formulas and symbols, which will be explained in a Word document.
